lnc-SAMD14-4 can regulate expression of the COL1A1 and COL1A2 in human chondrocytes

Zhang Haibin 1
Chen Cheng jc163yy@163.com 1
Cui Yinghong 2
Li Yuqing 3
Wang Zhaojun 4
Mao Xinzhan 5
Dou Pengcheng 5
Li Yihan 5
Ma Chi 6
1 Department of Orthopedics, The NO.921 Hospital of the People’s Liberation Army Joint Support Force, The Second Affiliated Hospital of Hunan Normal University , Changsha , Hunan , China
2 Department of Pharmaceutical Sciences, Hunan Normal University , changsha , Hunan , China
3 Department of Orthopedics, Changsha central hospital , Changsha , Hunan , China
4 Department of Traumatology, Shanxi Fenyang Hospital, The Fenyang Hospital of Shanxi Medical University , Fenyang , Shanxi , China
5 Department of Orthopedics, The Second Xiangya Hospital of Central South University , Changsha , Hunan , China
6 Department of Orthopedics, People’s Hospital of Xiangxi Autonomous Prefecture , Jishou , Hunan , China
Fortes Puri
Electronic publication date: 2019 Sep 2
Publication date: 2019
Volume: 7
Electronic Location ID: e7491
Received 2018 May 28; Accepted 2019 Jul 16
Copyright: ©2019 Zhang et al.
Copyright year: 2019
Copyright holder: Zhang et al.
License: This is an open access article distributed under the terms of the Creative Commons Attribution License, which permits unrestricted use, distribution, reproduction and adaptation in any medium and for any purpose provided that it is properly attributed. For attribution, the original author(s), title, publication source (PeerJ) and either DOI or URL of the article must be cited.
License URL: https://creativecommons.org/licenses/by/4.0/

Keywords: Long-coding ribonucleicacids, Microarray, Expression profile, Osteoarthritis

Funding: National Natural Science Foundation of China #81201432 81371997 Army Medical Research Subject of the 12th Five-Year-Plan fund #CWS11J275 Scientific Research Fund of Hunan Provincial Education Department #15B140 Hunan Province Health and Life Committee Scientific Research Project #B2016148 Military Medical Science and Technology Youth Training Program #2018JJ6033 Hunan Provincial Innovation Foundation for Postgraduates #CX2015B185 The study was supported by the National Natural Science Foundation of China (#81201432, 81371997), the Army Medical Research Subject of the 12th Five-Year-Plan fund (#CWS11J275), the Scientific Research Fund of Hunan Provincial Education Department (#15B140), the Hunan Province Health and Life Committee Scientific Research Project (#B2016148), the Military Medical Science and Technology Youth Training Program (#2018JJ6033), and the Hunan Provincial Innovation Foundation for Postgraduates (#CX2015B185). The funders had no role in study design, data collection and analysis, decision to publish, or preparation of the manuscript.

==============================
Osteoarthritis (OA) is the most common motor system disease in aging people, characterized by matrix degradation, chondrocyte death, and osteophyte formation. OA etiology is unclear, but long noncoding RNAs (lncRNAs) that participate in numerous pathological and physiological processes may be key regulators in the onset and development of OA. Because profiling of lncRNAs and their biological function in OA is not understood, we measured lncRNA and mRNA expression profiles using high-throughput microarray to study human knee OA. We identified 2,042 lncRNAs and 2,011 mRNAs that were significantly differentially expressed in OA compared to non-OA tissue (>2.0- or < − 2.0-fold change; p < 0.5), including 1,137 lncRNAs that were upregulated and 905 lncRNAs that were downregulated. Also, 1,386 mRNA were upregulated and 625 mRNAs were downregulated. QPCR was used to validate chip results. Gene Ontology analysis and the Kyoto Encyclopedia of Genes and Genomes was used to study the biological function enrichment of differentially expressed mRNA. Additionally, coding-non-coding gene co-expression (CNC) network construction was performed to explore the relevance of dysregulated lncRNAs and mRNAs. Finally, the gain/loss of function experiments of lnc-SAMD14-4 was implemented in IL-1β-treated human chondrocytes. In general, this study provides a preliminary database for further exploring lncRNA-related mechnisms in OA.

Introduction

Osteoarthritis (OA) is an emerging global health burden prevalent among aging populations (Nho et al., 2013; Loeser, 2013). OA is caused by cartilage matrix degradation, chondrocyte death, and osteophyte formation (Hayami et al., 2006; Carames et al., 2010). The pathology of OA is fairly well understood and multiple therapies exist to treat patient symptoms. These include corticosteroids and nonsteroidal anti-inflammatory drugs (Bohensky et al., 2009; Bellamy et al., 2006), experimental stem cell therapies to treat specific forms of OA, and biologics to address specific inflammatory mediators, such as cytokines. OA is also commonly treated with joint replacement surgery. However, there is continued desire to find new therapeutic targets to improve osteoarthritis treatment.

Recent studies have evaluated the gene regulation capabilities of long noncoding ribonucleic acids (lncRNAs) in OA development. LncRNAs are non-protein-coding transcripts (>200 nucleotides) that participate in numerous pathological and physiological processes (Guttman & Rinn, 2012; Wang & Chang, 2011; Palazzo & Lee, 2015). Some lncRNAs have been defined as key regulators in the onset and development of OA (Zhang et al., 2016a; Pearson et al., 2016; Su et al., 2015; Song et al., 2014; Liu et al., 2014; Steck et al., 2012). For instance, overexpression of HOTAIR may contribute to OA development by inducing chondrocyte apoptosis (Zhang et al., 2016a; Xing et al., 2014). Fu et al. (2015) used microarray and bioinformatics to identify 4,714 lncRNAs differentially expressed in knee cartilage from OA and non-OA patients. Additionally, Liu et al. (2014) used microarrays to identify 153 lncRNAs differentiallly expressed in OA patients and designated a specific lncRNA CIR as critical to degradation of the chondrocyte matrix. Differences in lncRNA expression are often attributed to physiological processes that arise in laboratories using high-throughput screening. To address the lack of information about the role of lncRNAs in the pathogenesis of OA, we used high-throughput microarray to measure differentially expressed lncRNAs and mRNAs from the human knee cartilage of OA patients and non-OA patients.

Materials and Methods

Specimen collection

A Kellgren-Lawrence (KL) system was used to measure OA severity in patients (Kellgren & Lawrence, 1957). OA cartilage (KL grades 3 or 4) was collected from 34 patients undergoing total knee joint replacement due to severe OA. Normal cartilage (KL grades 0) was collected from 19 patients without history of rheumatoid arthritis or OA (who were being treated for trauma, thromboangiities obliterans, osteosarcoma, or limb amputation). Informed consent was signed by all patients and additional informed consent was obtained for publication of subject age and sex. This study was approved by the Ethics Commission of the 163rd Central Hospital of the Chinese People’s Liberation Army (IRB. [2015] 001). Patient data appears in Table S1. Clinical specimens were divided for qPCR and for array analyses.

LncRNA and mRNA microarray analysis

Quality testing, sample labeling, and hybridization to human WT lncRNA microarray (Affymetrix) were conducted by Oebiotech Co (Shanghai, China) (Fig. S1); their detailed protocol can be found in previous publications (Lightfoot, 2016; Mueller et al., 2015). All RNA integrity number (RIN) values were >7, and 260:280 ratios were >1.7 (as measured by NanoDrop). The Genechip included 63,542 non-coding and 27,134 coding transcripts. Images were taken with an Affymetrix Scanner3000, and data normalization and probe filtering were conducted using an Affymetrix GeneChip Expression Console (version 4.0; Affymetrix). The lncRNA-mRNA expression between OA and non-OA groups was identified as a ≥2.0 fold-change (FC) for up-regulation or a ≤ − 2.0 fold-change for down-regulation, with p ≤ 0.05. Afterwards, GO and KEGG analyses were used to identify roles of differentially expressed mRNAs with respect to GO terms or pathways. Their detailed definitions can be found in previous publications (Huang et al., 2011; Tao et al., 2012).

A volcano plot and hierarchical clustering were created to show distinctions in gene expression patterns among samples. Finally, a coding-non-coding gene co-expression (CNC) network was drawn using Cytoscape software. The microarray data are available through the GEO database with the accession number GSE113825.

RNA extraction and quantitative PCR

A RNeasy Lipid Tissue Mini Kit (Qiagen, Hilden, Germany) was used to extract total RNA from tissues according to the manufacturer’s directions. A NanoDrop ND-1000 spectrophotometer was used to measure RNA quality and purity. Then SDS-PAGE was used to measure RNA integrity. Total RNA from samples was reverse transcribed into cDNA using an All-in-one First-Strand cDNA Synthesis Kit (GeneCopoeia Inc, Rockville, MD, USA) as directed by the user manual. qPCR primers were designed and synthesized at the Shanghai Sangon Company (Table S2); qPCR was conducted using AceQ qPCR SYBR Green Master Mix (Vazyme Biotech Co, Jiangsu Sheng, China) on a Bio-Rad CFX ConnectTM Real-Time PCR detection system (Bio-Rad, Munchen, Germany) using 10 µl SYBR Master Mix, 0.4 µl forward primer (10 µM), 0.4 µl reverse primer (10 µM), 0.4 µl ROX Reference Dye 1, two µl cDNA, and 6.8 µl MiliQ water. The qPCR reaction conditions were as follows: pre-denaturation at 95 °C for 5 min, followed by 95 °C for 10 s, and 60 °C for 30 s for 40 cycles, and a final extension at 95 °C for 15 s, 60 °C for 60 s, and 95 °C for 15 s. The 2−ΔΔCt method was employed to assess gene expression, as described in previous studies (Livak & Schmittgen, 2001). Each experiment was conducted in triplicate.

Cell culture and treatment

Human primary chondrocytes were procured from CHI Scientific, Inc (China). The cells were incubated in DMEM (Gibco, Waltham, MA, USA) medium supplemented with 14% FBS (Gibco, USA) in 6-well culture plates (6 × 105∕well). Cell cultures were cultivated at 37 °C in a 5% CO2 incubator and the medium was changed every three days. All studies were conducted using primary chondrocytes from passages 1-4. Cells were treated either with or without 10 ng/ml human recombinant interleukin-1 beta (IL-1 β) (Sigma-Aldrich, St Louis, MO, Germany) for 12 h, 24 h and 48 h.

Cell transfection

The overexpression vector, lnc-SAMD14-4(pcDNA3.1+-lnc-SAMD14-4), and blank vector, pcDNA3.1(pcDNA3.1+- vector), were designed by and purchased from GeneChem Co., Ltd. (Shanghai, China). The vectors were transfected into human primary chondrocytes using Lipofectamine 3000 (Invitrogen, Carlsbad, CA, USA). Three targeting siRNA, including si-lnc-SAMD14-4 (1#, 2# and 3#), and a control siRNA were designed and purchased from Ribo Biotechnology Co. Ltd. (Guangdong, China). These siRNA (100nM/well) were transfected into human primary chondrocytes to assess gain or loss of function of lnc-SAMD14-4.

Statistical analysis

Differences in expression of lncRNA-mRNA between OA and non-OA groups were assessed with a Student’s t test. GO and KEGG pathways were assessed using Fisher’s exact test. Other data were analyzed using one-way ANOVA and multiple groups were compared using Bonferroni’s method. GraphPad Prism 6.0 (GraphPad Software San Diego, CA) was applied for data analysis and drawing. Data are shown as means ± SD, and p < 0.05 was considered statistically significant.

Results

Analysis of differential expression among lncRNAs-mRNAs in OA

We found 2,042 lncRNAs differed significantly in expression levels between OA and non-OA tissue (Table S3). The most upregulated lncRNA was lnc-SAMD14-4 and the most downregulated lncRNA was lnc-MSMP-2. The 15 most dysregulated (either down or up) lncRNAs are listed in Table 1.

Table 1 Most regulated lncRNAs.

GeneSymbol	Regulation	Type	Fold change(abs)	P-value	
lnc-SAMD14-4	Up	Exonic	86.11047	0.00036	
lnc-MARCKS-7	Up	——-	80.51785	0.0000923	
lnc-SAMD14-5	Up	Exonic	76.39162	0.0000161	
lnc-AP3S2-2	Up	Exonic	57.44485	0.001265177	
lnc-CASD1-2	Up	Exonic	56.14244	0.001261528	
lnc-TSPAN13-4	Up	Exonic	50.915787	0.00000633	
lnc-PPM1N-1	Up	Exonic	47.80361	0.009312962	
lnc-TRPC4-2	Up	Exonic	47.43022	0.009825662	
lnc-C1orf196-1	Up	Linc	40.125366	0.0000202	
lnc-SMAD5-7	Up	Exonic	39.585285	0.022567188	
lnc-TSPAN13-5	Up	Exonic	37.4391	0.00000077	
lnc-SFRP1-2	Up	Exonic	36.96935	0.001035282	
lnc-HJURP-12	Up	Exonic	36.914413	0.000862	
lnc-TEX2-2	Up	Exonic	33.92366	0.000572	
lnc-MS4A6E-1	Up	Exonic	31.000843	0.001087416	
lnc-MSMP-2	Down	Exonic	22.78652	0.030858	
lnc-NPVF-4	Down	Linc	18.66038	0.0000169	
lnc-CBLL1-4	Down	Exonic	14.84985	0.022533	
MLLT4-AS1	Down	——-	12.8248	0.002052	
lnc-NPVF-10	Down	——-	12.68418	0.0000278	
lnc-ASZ1-3	Down	Exonic	11.32834	0.000425	
SAMD12	Down	——-	9.023141	0.006789	
lnc-NPVF-5	Down	——-	9.018275	0.000095	
lnc-IRGM-3	Down	Exonic	8.909008	0.001227	
lnc-ERCC5-4	Down	Linc	8.883492	0.000686	
lnc-SSR3-6	Down		8.876818	0.000349	
lnc-DLGAP1-9	Down	Exonic	8.776601	0.001021	
lnc-SSR3-5	Down	Antisense	8.354682	0.000208	
lnc-CHI3L1-1	Down	Exonic	8.350381	0.011126	
lnc-OXNAD1-6	Down	Antisense	8.14148	0.0000128	

mRNA expression data revealed dysregulation of 2,011 mRNAs in OA cartilage tissue, as compared to non-OA cartilage tissue; it was found that 1,386 mRNA were upregulated and 625 mRNAs were downregulated (Table S4). Among these mRNAs, the most upregulated protein coding RNA was MMP-13, which is responsible for regulating articular cartilage degradation (Fosang et al., 1996). MYOC was the most strongly down-regulated. The top 15 upregulated and downregulated mRNAs appear in Table 2. Data on differentially expressed genes (including lncRNA and mRNA) appear in Figs. 1A–1C.

Table 2 Most regulated mRNAs in this study.

GeneSymbol	Regulation	Fold change(abs)	P-value	
MMP13	Up	272.535	0.000564	
SERPINF1	Up	191.17508	0.000000141	
COL1A1	Up	175.43033	0.0000199	
POSTN	Up	156.12422	0.008303139	
LRRC15	Up	105.57993	0.000129	
COL1A2	Up	85.32694	0.000776	
THBS2	Up	74.47372	0.002686227	
FNDC1	Up	72.167336	0.001257106	
RP11-429G19.3	Up	69.320656	0.000105	
MS4A4A	Up	64.19533	0.000564	
FPR3	Up	56.89943	0.0000552	
MSR1	Up	53.758953	0.000178	
AHR	Up	52.798218	0.0000266	
TNFAIP6	Up	51.144848	0.018054951	
PLXDC1	Up	51.0092	0.00043	
MYOC	Down	49.693558	0.008422094	
PAK3	Down	35.395626	0.014492353	
APOD	Down	26.913952	0.011042964	
MSMP	Down	24.774714	0.036745384	
SLC26A4	Down	20.350153	0.024579711	
AC003090.1	Down	19.897524	0.0001	
CYP4F11	Down	12.0151615	0.001603253	
CTTNBP2	Down	11.738498	0.000434	
ABCA5	Down	10.876956	0.000144	
ZMAT1	Down	9.586397	0.000137	

Figure 1 Volcano plot and hierarchical clustering of lncRNA and mRNA differential expression profiles between non-OA and OA groups for 10 cartilage tissues.

(A) Volcano plot of dysregulated lncRNA and mRNA (dots denotes gene). Grey spots represent p > 0.05; green spots represent p < 0.05 and fold-change <2. Red spots represent up-regulated genes (p < 0.05 and fold-change >2.0), and blue spots represent up-regulated genes (p < 0.05 and fold-change < − 2.0). Heatmap of differentially expressed lncRNA (B) and mRNA (C). Colors denote relative expression: red represents high relative expression; green represents low relative expression.

Gene Ontology (GO) analysis and Kyoto Encyclopedia of Genes and Genomes (KEGG) analysis of dysregulated mRNAs

The most enriched GOs targeted by upregulated mRNAs were extracellular matrix organization (GO:0030198), extracellular exosome (GO:0070062), and collagen binding (GO:0005518) in biological process, cellular process, and molecular function (Figs. 2A–2C, Tables S5–S7). Protein stabilization (GO:0050821), cilium (GO:0005929), and heparan sulfate proteoglycan binding (GO:0043395) were the most enriched GO pathways associated with downregulated transcripts (Figs. 2D–2F, Tables S8–S10).

Figure 2 Go analysis of abnormal mRNAs between OA and non-OA groups.

Enriched Go terms targeted by up-regulated mRNAs (A–C) and down-regulated mRNAs (D–F) according to biological processes, cell components, and molecular function (p < 0.05 as statistically significant according to a Fisher’s exact test).

We identified 27 pathways with upregulated mRNAs and the most enriched pathway was cytochrome P450 (Table S11). Additionally, 94 pathways corresponded to downregulated mRNAs, according to KEGG pathway analysis, and the top enriched pathway was focal adhesion (Table S12). The top 20 pathways enriched by downregulated and upregulated transcripts appear in Figs 3A and 3B.

Figure 3 KEGG pathway analysis of abnormal mRNAs between OA and non-OA groups.

The top 20 pathways targeted by down-regulated (A) and up-regulated (B) transcripts (p < 0.05 was statistically significant according to a Fisher’s exact test).

LncRNA-mRNA co-expression network

Co-expression networks are of biological interest because co-expressed genes are controlled by the same transcriptional regulatory program, are functionally related, or are members of the same pathway or protein complex (Weirauch, 2011; Stuart et al., 2003). It has been confirmed that apoptosis is involved in OA pathogenesis (Zhang et al., 2016b; Hwang & Kim, 2015; Andrew et al., 2007; Kim & Blanco, 2007), and that the PI3K-Akt signaling pathway is important for apoptosis (Franke et al., 2003). We analyzed the 74 dysregulated mRNAs found to be enriched in the PI3K-Akt signaling pathway (Table S13) and identified 1,974 correlations for lncRNA-mRNA interaction pairs that satisfy p < 0.01 and Pearson correlation coefficients>0.95 (Table S14). Then, the top 500 of 1,974 lncRNA-mRNA pairs were used to draw the network (Table S15). Of the top 500 lncRNA-mRNA pairs, which included 57 mRNAs and 363 lncRNAs, 362 pairs had a positive association; the rest were negatively associated. It appears that a single mRNA may pair with multiple lncRNAs while most lncRNAs may only pair with a single mRNA (Fig. 4). Some lncRNA-mRNA pairs included the most dysregulated mRNA and lncRNA. For example, COL1A1 (mRNA, FC = 175.43) and COL1A2 (mRNA, FC = 85.32) were strongly associated with lnc-SAMD14-4 (lncRNA, FC = 86.11). All lncRNA-mRNA co-expressed pairs had Pearson correlation coefficients>0.98

Figure 4 LncRNA-mRNA co-expression network.

Green node represents mRNA; red node represents lncRNA. Line indicates a co-expression relationship between lncRNA-mRNA. The node degree is indicated by the node size.

Verification of lncRNAs and mRNAs expression by qRT-PCR

To confirm microarray profiling analysis, four lncRNAs (lnc-SAMD14-4, lnc-MARCKS-7, lnc-MSMP-2 and lnc-NPVF-4) and four mRNAs (SERPINF1, COL1A1, MYOC and PAK3) were selected and measured using qRT-PCR in 53 clinical specimens (34 OA and 19 non-OA specimens). The results from the qRT-PCR agreed with chip profiling analysis (Fig. 5).

Figure 5 qPCR and microarray data.

Values are means + SD (n = 53), and * p < 0.05 was statistically significant compared to the non-OA group.

Prediction of novel lncRNA targeting

We studied the most dysregulated lncRNA, lnc-SAMD14-4, for its potential function in OA development. To analyze the gene coding-potential of lnc-SAMD14-4, the open-reading frames (ORFs) and codon-substitution frequency (CSF) were measured according to former research protocol (Liu et al., 2014; Qu et al., 2016). The ORF Finder (National Center for Biotechnology Information) failed to predict ORFs more than 300 nt. Furthermore, BLAST search of all the positive-sense strand ORFs, which are marked by “ +” in Fig. S2, yielded no highly homologous protein among short peptides. Alternatively, PhyloCSF analysis of lnc-SAMD14-4 was negative (Fig. S3). These results confirmed that lnc-SAMD14-4 does not encode any functional protein product.

COL1A1, COL1A2 and lnc-SAMD14-4 were highly expressed in the IL-1 β-treated human primary chondrocytes

According to our co-expression network analysis, COL1A1 and COL1A2 were strongly associated with lnc-SAMD14-4. Thus, we speculated that lnc-SAMD14-4 may contribute to OA by interacting with COL1A1 and COL1A2. Previous studies have used human primary chondrocytes treated with IL-1 β-treated to model the process of human osteoarthritis (Fu et al., 2015; Goldring et al., 1988). We determined expression levels of COL1A1, COL1A2 and lnc-SAMD14-4 by qPCR. We found that COL1A1 and COL1A2 were upregulated after human chondrocytes were treated with IL-1 β for 24 h and 48 h. Furthermore, upregulation of lnc-SAMD14-4 with IL-1 β treatment was time-dependent (Fig. 6). These results suggest a positive correlation between lnc-SAMD14-4 and COL1A1, COL1A2 expression induced by IL-1 β treatment. Hence, human primary chondrocytes were treated either with or without IL-1 β for 48 h in the follow-up experiments.

Figure 6 The mRNA levels of COL1A1 (A), COL1A2 (B) and lnc-SAMD14-4 (C) were detected by QPCR in the IL-1 β-treated human primary chondrocytes.

The mRNA expression was normalized to GAPDH expression and conversion by 2−ΔΔCt. Values are means + SD (n = 3). Data were analyzed using one-way ANOVA followed by Dunnett-ttest.

lnc-SAMD14-4 was involved in IL-1 β-induced COL1A1 and COL1A2 expression in human primary chondrocytes

To further explore the function of lnc-SAMD14-4 in eliciting IL-1 β-induced COL1A1 and COL1A2 expression, we performed gain/loss of function experiments. We confirmed that lnc-SAMD14-4 expression in human primary chondrocytes increased after pcDNA3.1+- lnc-SAMD14-4 transfection (Fig. S4). Additionally, as shown in Fig. 7, lnc-SAMD14-4 overexpression promoted expression of COL1A1 and COL1A2 in the human primary chondrocytes, regardless of treatment with IL-1 β (p < 0.05).

Figure 7 Effects of lnc-SAMD14-4 overexpression on the relative expression levels of COL1A1 (A), COL1A2 (B) in IL-1 β-treated human primary chondrocytes detected by QPCR.

Data were analyzed using one-way ANOVA followed by Dunnett-ttest.

To explore the knockdown effects of lnc-SAMD14-4 on COL1A1 and COL1A2, we designed and transfected three independent si-lnc-SAMD14-4 (1#, 2# and 3#) in human primary chondrocytes. The strongest silencing effects were observed in si-lnc-SAMD14-4 (2#), which was used for silencing in subsequent experiments (Fig. S5). The knockdown of lnc-SAMD14-4 significantly decreased the expression of COL1A1 and COL1A2 in human primary chondrocytes, regardless of treatment with IL-1 β-treatment (Fig. 8).

Figure 8 Effects of lnc-SAMD14-4 suppression on the relative expression levels of COL1A1 (A), COL1A2 (B) in IL-1 β-treated human primary chondrocytes detected by QPCR.

Data were analyzed using one-way ANOVA followed by Dunnett-ttest.

Discussion

To our knowledge, there has been some study focusing on the lncRNA expression profiles in osteoarthritis (OA). For instance, Fu et al. identified 4,714 lncRNAs and Liu et al. identified 152 lncRNAs that were differentially expressed in OA cartilage as compared to non-OA cartilage Pearson’s group treated primary chondrocytes isolated from hip articular cartilage with IL-1 β to stimulate OA. This led them to identify lncRNAs related to the inflammatory response in human hip OA cartilage (Pearson et al., 2016; Fu et al., 2015). However, there was much left to learn regarding the roles of lncRNAs in osteoarthritis.

In the present study, we analyzed human knee cartilage from OA and non-OA patients by measuring expression profiles of lncRNAs and mRNAs using high-throughput microarray and bioinformatics. For this, we used a standards filter of FC ≥2.0, p < 0.05. An important point to note is that our work used Affymetrix and previous studies used Arraystar, which is a different platform. This was problematic when comparing lncRNA data.

Using these chips, we identified 2,042 dysregulated lncRNAs and 2,011 differentially expressed mRNAs in OA samples as compared to non-OA tissues. These included COL1A1, COL1A2, MMP-13, MMP-9, and TNFAIP6, which are associated with cartilage biology and OA (Freyria & Mallein-Gerin, 2012; Attur et al., 2015; Al-Sabah et al., 2016; Roberts et al., 2009; Snelling et al., 2014). Excluding MMP13 mRNA from the study (which has already been linked to chondrocyte biological processes and osteoarthritis (Fosang et al., 1996)), studies with the top four dysregulated lncRNAs and mRNAs verified that qRT-PCR data for 53 specimens (34 OA and 19 normal specimens) were consistent with chip analysis.

Next, GO analysis and pathway analysis analyzed the biological function enrichment of differentially expressed mRNA. For instance, the most enriched GO terms targeted by up-regulated mRNAs were extracellular matrix organization (GO:0030198), extracellular exosome (GO:0070062), and collagen binding (GO:0005518) in biological processes, cellular processes, and molecular function. Functions of the extracellular matrix and collagen binding have been accepted as important to the onset and progression of OA (Svensson et al., 2001; Blaney Davidson, van Caam & van der Kraan, 2017).

A lncRNA-mRNA co-expression network was constructed to identify associations between lncRNAs and mRNAs. The fundamental principle of co-expression network analysis is that opposite or identical expression pattern of mRNAs and lncRNAs indicate some interaction between the lncRNA-mRNA. We noticed that most dysregulated lncRNA or mRNA made lncRNA-mRNA pairs. Among these, lnc-SAMD14-4 (lncRNA, FC = 86.11) was strongly associated with COL1A1 (mRNA, FC = 175.43) and COL1A2 (mRNA, FC = 85.32). Type I collagen is a triple helix consisting of two pro-alpha1(I) chains encoded by two COL1A1 genes and one pro-alpha2(I) chain encoded by one COL1A2 gene. Type I collagen is the most abundant collagen of the human body present in scar tissue and it is also regarded as the end product when tissue heals (Junqueira & Uchôa, 2013). It is reported that Type I collagen is rare in normal chondrocytes, however, higher levels of Type I collagen was detected in osteoarthritis cartilage, particularly in late stages of osteoarthritis (Miosge et al., 2004).

It is reasonable to suggest that lnc-SAMD14-4 may promote expression of the COL1A1 gene and COL1A2 gene, thereby contributing to the pathogenesis of OA. To verify this hypothesis, Il-1 β-treated human chondrocytes were used to imitate the genetic change associated with osteoarthritis development, as uncovered in previous studies (Fu et al., 2015; Shijin Lu & Cheng, 2015). Our study suggested a positive correlation between expression of lnc-SAMD14-4 and COL1A1 gene and COL1A2 gene in Il-1 β-treated human chondrocytes. This is consistent with expression patterns found in cartilage from osteoarthritis patients. Thus, lnc-SAMD14-4 may play a key role in the pathogenesis of osteoarthritis by promoting the expression of the COL1A1 and COL1A2 genes.

Age is the highest risk factor for the pathogenesis of osteoarthritis. This link between aging and OA is complex, but there are some common physical mechanisms in aging and OA. One limitation of this study is the lack of age-matched patients. In clinical practice, the age of OA patients was usually older than the age of non-OA patients. However, some of the most dysregulated mRNAs in our study, including MMP-13 and POSTN, were OA-related (Brophy et al., 2017; Davidson et al., 2006; Karlsson et al., 2010), which suggests that the differences observed in our arrays were at least in part a consequence of OA. Therefore, further research should be performed with age-matched patients or a more ideal model in the future.

Conclusion

Taken together, the Affymetrix chips and bioinformatics methods were combined to explore differential expression of lncRNAs from OA articular cartilage and non-OA articular cartilage. This study revealed that lnc-SAMD14-4 may upregulate the COL1A1 and COL1A2 genes, contributing to the pathogenesis of OA. This provides a potential therapeutic target for treatment.

Supplemental Information

Supplemental Information 1 The MIAME checklist

Click here for additional data file.

Figure S1 The quality control of total RNA

(A) the SDS-PAGE gels of total RNA. 28S ribosomal RNA and 18S ribosomal RNA bands appeared in all tissue samples (B) The electrophoresis pattern of SDS-PAGE of total RNA. (C) the analysis of SDS-PAGE of total RNA, RNA Integrity Number(RIN) >7 represent the quality of RNA can satisfy the study requirements.

Click here for additional data file.

Figure S2 Prediction of putative proteins encoded by lnc-SAMD14-4 using ORF Finder

Click here for additional data file.

Figure S3 The codon substitution frequency scores(CSF) of lnc-SAMD14-4

Click here for additional data file.

Figure S4 QPCR analysis confirmed increased lnc-SAMD14-4 expression in human primary chondrocytes after pcDNA3.1 +-lnc-SAMD14-4 transfection

QPCR analysis confirmed increased lnc-SAMD14-4 expression in human primary chondrocytes after pcDNA 3.1+-lnc-SAMD14-4 transfection.

Click here for additional data file.

Figure S5 The effect of three independent specifically targeting siRNA against lnc-SAMD14-4 was measured by QPCR

The strongest inhibitory result was achieved by lnc-SAMD14-4(2#)

Click here for additional data file.

Table S1 Patient data

Click here for additional data file.

Table S2 qPCR primers used in this study

Click here for additional data file.

Table S3 Differentially expressed lncRNA in OA and non-OA groups

Click here for additional data file.

Table S4 Differentially expressed mRNA in OA and non-OA groups

Click here for additional data file.

Table S5 Biological processes found with Go enrichment analysis for up-regulated mRNA in OA and non-OA groups

Click here for additional data file.

Table S6 The cellular process of Go enrichment analysis for up-regulated mRNA in OA and non-OA groups

Click here for additional data file.

Table S7 Molecular functions for Go enrichment analysis of up-regulated mRNA in OA and non-OA groups

Click here for additional data file.

Table S8 Biological processes for GO enrichment analysis of down-regulated mRNA in OA and non-OA groups

Click here for additional data file.

Table S9 Cellular process for GO enrichment analysis of down-regulated mRNA in OA and non-OA groups

Click here for additional data file.

Table S10 Molecular functions for GO enrichment analysis of down-regulated mRNA in OA and non-OA groups

Click here for additional data file.

Table S11 Pathway analysis of up-regulated mRNA in OA and non-OA groups

Click here for additional data file.

Table S12 Pathway analysis of down-regulated mRNA in OA and non-OA groups

Click here for additional data file.

Table S13 Dysregulated mRNAs enriched in the PI3K-Akt signal pathway

Click here for additional data file.

Table S14 The lncRNA-mRNA interaction pairs based on the PI3K-Akt signal pathway

Click here for additional data file.

Table S15 Top 500 correlations of lncRNA-mRNA interaction pairs based on the PI3K-Akt signal pathway

Click here for additional data file.

Additional Information and Declarations

Competing Interests

Author Contributions

Human Ethics

Microarray Data Deposition

Data Availability

The authors declare there are no competing interests.

Haibin Zhang conceived and designed the experiments, performed the experiments, analyzed the data, contributed reagents/materials/analysis tools, prepared figures and/or tables, authored or reviewed drafts of the paper, approved the final draft.

Cheng Chen and Yinghong Cui conceived and designed the experiments, performed the experiments, contributed reagents/materials/analysis tools, prepared figures and/or tables, authored or reviewed drafts of the paper, approved the final draft.

Yuqing Li performed the experiments, contributed reagents/materials/analysis tools, authored or reviewed drafts of the paper.

Zhaojun Wang performed the experiments, contributed reagents/materials/analysis tools.

Xinzhan Mao performed the experiments.

Pengcheng Dou, Yihan Li and Chi Ma performed the experiments, provided the study materials.

The following information was supplied relating to ethical approvals (i.e., approving body and any reference numbers):

This study was approved by the Ethics commission of the 163rd Central Hospital of the Chinese People’s Liberation Army (IRB. [2015] 001).

The following information was supplied regarding the deposition of microarray data:

GEO: GSE113825.

The following information was supplied regarding data availability:

The raw data are included in the figures, tables, and Supplemental Tables.

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
