# Peer review of "lnc-SAMD14-4 can regulate expression of the COL1A1 and COL1A2 in human chondrocytes"

_PeerJ, doi:10.7717/peerj.7491_

## Round 0.1 · original submission · Major Revisions

In general, my opinion agrees with that of the reviewers. The manuscript requires a stronger focus and a real conclusion. The results need an anlysis that reachs a biological meaning. By comparing the data with that of the other reported studies, more important information should be obtained. Validation of the results should be more intense. Validation should be enlarged to include those candidates that are reproducibly deregulated in several studies and those with a Fold Change closer to 2. All the comments raised by the reviewers should be carefully addressed. Finally, the ethical approval needs to be in English.

Reviewer 1 ·

Basic reporting

There is not a structured abstract as peerJ guide recommend. Abstract should organized in background, methods, results and discussion.
Authors have used an English langue editor as a result the manuscript is clear and well writing. However, some sentences in the conclusion are confusing and not well written.
References should be updated if the authors follow this reviewer´s recommendations.
Figures are well labeled with good quality and well described.
Raw data is supplied.

Experimental design

The study might be considered inside the scope of peerJ.
The objectives are not well defined. It is not clear if they want to compare the efficacy of two different microarrays or they really want to discover expression changes in OA patients.
Techniques are well performed and controlled.
There are some gaps in methods that are further commented.

Validity of the findings

Even number of patients is low, statistical analysis is well performed although a an explanation is necessary about the use of ANOVA test

Authors compare their findings with others reported by Fu et al. The authors should extend their comparisons in discussion to other papers related to this topic. Here I suggest some papers that authors might use for this comparison.
Cell Death Dis. 2018 Jun 15;9(7):711. doi: 10.1038/s41419-018-0746-z,
J Cell Biochem. 2018 Jun 12. doi: 10.1002/jcb.27057.
Eur Rev Med Pharmacol Sci. 2018 May;22(10):2966-2972.
Biomed Pharmacother. 2018 Aug;104:825-831. doi: 10.1016/j.biopha.2018.04.124. Epub 2018 Apr 25.10.26355/eurrev_201805_15051.
Biochem Biophys Res Commun. 2018 Jun 7;500(3):658-664. doi: 10.1016/j.bbrc.2018.04.130. Epub 2018 Apr 24.
Cell Physiol Biochem. 2018;45(3):1241-1251. doi: 10.1159/000487455. Epub 2018 Feb 9.
Cell Physiol Biochem. 2018;45(2):832-843. doi: 10.1159/000487175. Epub 2018 Jan 31.

There are few comments in the discussion about the biological importance of the findings and how they might be related to advance in discovering new treatment for OA.

Conclusion is poor and confusing, it needs to be rewritten. It is not clear what the authors conclude and the importance of the study.

Additional comments

Table 1 must be moved to supplementary material

Patient data related to age and sex should be shown. Is it possible that there were any bias in sex or age between OA and non-OA groups?

There are little information about how specimens were treated to perform lncR microarray.

Analysis of qPCR must cited the work of Livat & Schmittgen 2001 (http://www.gene-quantification.net/livak-2001.pdf)

What are the other data analyzed with ANOVA?

A reference is required in this sentence “In previous work, the most enriched GOs targeted by up-regulated mRNAs were anatomical structure development, cell periphery, and lipid binding in biological processes, cellular processes, and molecular function, respectively.

Reviewer 2 ·

Basic reporting

The reporting is overall clear and written in professional English, but I have found quite a few mistakes listed below:
-L43: In genel > is it “in general”?
-L74: informed consent was signed all patients > informed consent was signed by all patients
-L97: according the manual > according to the manual
-L105: were assess > were assessed
-L106: a multiple groups > multiple groups
-L110: “Analysis Differentially expression”: should be rephrased
-L116: “this regulated articular cartilage degradation”: this should be rephrased, as it gives the impression you have confirmed that MMP13 up-regulation had an effect on cartilage degradation.
-L117: “MYOC was minimally expressed”: do you mean “MYOC was the most strongly down-regulated”?
-L119: “abnormal mRNAs”: what do you mean by “abnormal”?
-L133: “it is possible…mRNA”: there is no verb
-L144: was correlates > correlates
-L145: “top mRNA and lncRNA”: what does this mean?
-L146~148: this sentence is also difficult to understand.
-L172: “we identified 23 common …between 2 studies”: this sentence is also difficult to understand.
-L201: their the top > their top


The references seem adequate.


The article is properly structured with all the necessary figures and tables. However, why do you show the volcano plot with the lncRNAs and mRNAs combined, whereas the heatmaps show lncRNAs and mRNAs separately? Furthermore, line 160: “meaning that 244 were common for both studies”: I don’t understand where this figure (244) comes from. This should be better explained either in the main text or with the figure/table legends.

Experimental design

The research presented is within the aims and scope of the journal.

The research question is relevant and meaningful.

The investigation is performed to a high technical standard.

The methods, however, are sometimes confusing, especially regarding the microarrays used: were the chips from Arraystar or Affymetrix? Also, line 170, what is a “specification chip”?
How were the GO and KEGG analyses and the hierarchical clustering carried out?
Also, no data about the purity of the RNA is shown (SDS-PAGE); could you provide these in a supplementary figure?

Validity of the findings

The major strength of this article is, to me, the large number of samples used for the study. However, the results are very different from previously reported studies on microarrays using similar samples, so it is difficult to judge whether the results are robust or not. For instance, line 176~178: “In addition…our work (FC=6.28)”: does this mean there is only one gene that is similarly up-regulated in all three studies (ENST00000426074)? If very little overlapping results can be found between the three studies, how can this be explained? This is not discussed in the discussion. Did the other studies use less samples? Were the samples used treated differently? etc...

Also, PCR was only carried out on 8 genes, which were furthermore the most strongly modulated genes; is this sufficient to show the validity of the microarray results?

Regarding statistics, why was the SEM used instead of SD (standard deviation)?

Regarding the conclusions, only very little is said about the modulated lncRNAs (MMP-13, MMP-9, and TNFAIP6 are mentioned but those are mRNAs). As the paper is originally about lncRNAs in osteoarthritis, could you provide some discussion of the modulated lncRNAs identified in this study?

---

## Round 0.2 · Major Revisions

While one reviewer seems satisficed, the second reviewer has raised major and minor issues that deserve a proper answer. Proper validation should be done with age matched patients. The effect of lnc-SAMD14-4 on COL1A1 and COL1A2 should also be shown in IL-1b-treated cells. The relevance of COL1A1 and COL1A2 expression in OA has to be supported.

Reviewer 1 ·

Basic reporting

The manuscript meets the criteria of the journal

Experimental design

The experimental desing is acceptable.

Validity of the findings

Findings in the manuscript are admisable

Additional comments

The authors have satisfactorily addressed this reviewers comments

Reviewer 2 ·

Basic reporting

The authors say they have used an “English langue editor”; however, many mistakes remain, especially in the parts that have been newly added; I believe the editing has not been carried out properly by a native speaker and strongly recommend this should be done to improve the readability of the article. Here are a few corrections for the abstract:
P7, l31: “Because…not understood”: there is no independent clause
P7, l32: “a new lncRNA-lnc-SAMD14-4 and…” > “a new lncRNA (lnc-SAMD14-4) and…”
P7, l32: “potential ability”: do you mean “potential involvement”?
P7, l34: “The human cartilage tissues” > “Human cartilage tissues”
P7, l36: “related biological modules”: what does this mean? (“module” does not appear again in the text)
P7, l37: “The co-expression analysis … between the lncRNA-mRNA couple associated with PI3K-Akt pathway”: this is not very clear either; do you mean “Co-expression analysis was used to predict interactions between lncRNAs and mRNAs associated with the PI3K-Akt pathway”?
P7, l38: “the expression pattern of lncRNA and mRNA” should probably be “the expression pattern of some modulated lncRNAs and mRNAs”
P7, l39: “to explored” > “to explore”
P7, l42: “total 2042 lncRNAs were detected significantly different expression in OA” > “a total of 2042 lncRNAs were detected with a significantly different expression in OA”
P7, l43: “Total 2011 mRNA was detected significantly different expression in OA” > “A total of 2011 mRNAs were detected with a significantly different expression in OA”
P7, l44: “Go analysis” should probably be “GO analysis”
P7, l44: “all differentially expressed gene on pathogenesis” > “all differentially expressed genes on the pathogenesis”
P7, l44-46: there are 2 verbs in the sentence; maybe “corresponded” should be “corresponding”?
P8, l47: “mediated COL1A1 and COL1A2 expression” should probably be “mediated the COL1A1 and COL1A2 up-regulation”
P8, l49: “mechnisms” > “mechanisms”
In the Methods section,
P10, l84 and l90; “definition”; does that mean “protocol”?
P11, l105: “as previous study” > “as in previous studies”
P13, l155: “including” should probably be “included”


The references seem, overall, adequate. However, P16, l208-209: “COL1A1… which are associated with cartilage biology and OA”: one or two references should be given here.


The article is properly structured with all the necessary figures and tables. The raw data has been shared on the GEO database.
Regarding the figures, however,
- in Figure 3 legend: H and I should probably be A and B.
- legends of Fig6 and Fig7 should clearly state what A and B are.
- fig4 and P13, l156-157: one mRNA does seem to correlate with many lncRNAs but most lncRNAs only seem to correlate with one mRNA, so this sentence should be modified accordingly or better explained. Furthermore, the legend of Fig4 does not explain why the nodes have different sizes.
- finally, P10, l84: the contents of Supplement (or Supplementary?) figure 1 should be at least briefly mentioned

Experimental design

The research presented is within the aims and scope of the journal.

The research question is relevant and meaningful.

The investigation is performed to a high technical standard.

The methods, however, are still somewhat lacking:
- The authors mention that OA samples were taken from 19 patients and normal tissue from 11; however, the list of patients contains 53 patients; could this be explained?
- P14, l164: the main text states that 53 clinical specimens were used for PCR, but the legend of Fig5 mentions n=3; why is that?
- P11, l114-116: some more details regarding the transfection experiments should be given: how was the expression vector pcDNA3.1+-lnc-SAMD14-4 obtained? How many cells were seeded or transfected? With how much vector? etc

Validity of the findings

The previous version of the manuscript did not discuss the modulated lncRNAs identified in this study, but the revised version focuses on lncSAMD14-4 and its possible role in modulating COL1A1 and COL1A2, which has brought some improvement to the manuscript.

However, one of my main concerns, which is also completely lacking from the discussion, is the age of the OA and control groups: the mean age of the 5 OA patients used for microarray is 63 years; the mean age of the 5 control patients used for microarray is 28.8 years; are the differences in lncRNA and mRNA expression detected by microarray the result of OA or just old age? As the age difference could account for some or most of the differences observed (Loeser et al., Arthritis Rheum, 2012; Peffers et al., Arthritis Res Ther, 2013; Adapala et al., Gene, 2017), this should be discussed.

Furthermore, some conclusions seem to lack supporting evidence:
- P15, l189-190: it isn’t clear whether lnc-SAMD14-4 overexpression induces a significant increase in COL1A1 and COL1A2 expression in IL-1b-treated cells; the * only shows a difference compared to the control group; is there a significant difference between the IL-1b and IL-1b + lnc-SAMD14-4 group? Furthermore, how much was the lnc-SAMD14-4 expression increased in the transfected cells?
- P15, l193: the effect of the knock-down of lnc-SAMD14-4 by si-lnc-SAMD14-4(2#) should be shown (it generally is). As mentioned in the previous comment, the * only shows a difference compared to the control group; is there a significant difference between the IL-1b and IL-1b + si-lnc-SAMD14-4 group? What about between the siRNA (I suppose it’s the control siRNA group?) and si-lnc-SAMD14-4 group?
- P15, l195: “These data…osteoarthritis”: the only clear difference in COL1A1-2 expression is between the vector group and the lnc-SAMD14-4 group (unless further statistical analysis shows other significant differences); furthermore, the link between osteoarthritis and COL1A1 and COL1A2 expression has not been supported (by experimental results or references to other studies), so this sentence is, I believe, an overstatement.


Finally, the study does not compare its findings with similar reports on lncRNA expression in OA. On p16, l205-206: “Because…problematic”: although the platforms are different, I believe a comparison of the lists of modulated lncRNAs should be possible; if not, why could this not been done? If a comparison is possible, are there any overlaps between the current and previous studies and what are the main differences between the current and previous studies (especially Fu et al. and Liu et al.)?

---

## Round 0.3 · Major Revisions

I have been unable to re-evaluate your manuscript because the rebuttal letter corresponds to manuscript entitled“Pioglitazone Inhibits Advanced Glycation End Product-Induced Matrix Metalloproteinases and Apoptosis by Suppressing the Activation of MAPK and NF-κB”(ID:APPT-D-16-00111 )

Please, attach the correct rebuttal letter

#

---

## Round 0.4 · Minor Revisions

Please, be sure to correct the English in the revise version and to address all minor issues raised by the reviewer

Reviewer 2 ·

Basic reporting

The reporting is overall clear and written mostly in professional English, but I have still found quite a few mistakes. I appreciate that my English corrections for the abstract were taken into account, but the whole manuscript should have been checked by a native English speaker or a professional editing service before submitting the manuscript (as is usually the case).

The references seem adequate.

The article is properly structured with all the necessary figures and tables.

The article is self-contained with relevant results to hypotheses.

Experimental design

The research presented is within the aims and scope of the journal.

The research question is relevant and meaningful.

The investigation is performed to a high technical standard.

The methods are described with sufficient detail.

Validity of the findings

My main concern regarding the age difference between the control and OA groups has been mentioned in the discussion. I understand that getting age-matched controls for such a study is very challenging, so, as long as this limitation is clearly mentioned in the discussion, I can accept the results (knowing that I will have to be cautious with the data).

All underlying data have been provided. Controls have been added to figures 7 and 8 as required.

Conclusions are well stated, although, on L243-245: “most dysregulated… were OA-related”: this is probably an overstatement: just because a couple of strongly regulated genes are OA-related does not mean that “many of the dysregulated lncRNAs and mRNAs” are also OA-related; I suggest instead “… which suggests that the differences observed in our arrays were at least in part a consequence of OA”. (Though the authors mention that they “have contribute other paper about common dysregulated lncRNA between our Affymetrix chips work with Fu’s group Arraystar work”, a proper comparison between the current microarray data and previously published data would have been helpful to confirm that the gene modulations observed here were indeed OA-related.)

Additional comments

I appreciate that most of my previous concerns have been addressed.
I still have a few comments and questions:

The whole manuscript will have to be checked again by a native English speaker or a professional editing service before submitting the manuscript (as is usually the case...); there are still too many mistakes for me to point every one out (e.g. l112: "All study were" should be "All studies were", etc.).

L243-245: “most dysregulated… were OA-related”: this is probably an overstatement: just because a couple of strongly regulated genes are OA-related does not mean that “many of the dysregulated lncRNAs and mRNAs” are also OA-related; I suggest “… which suggests that the differences observed in our arrays were at least in part a consequence of OA”. (Though the authors mention that they “have contribute other paper about common dysregulated lncRNA between our Affymetrix chips work with Fu’s group Arraystar work”, a proper comparison between the current microarray data and previously published data would have been helpful to confirm that the gene modulations observed here were indeed OA-related.).

There are too many significant digits (in the main text and in tables). For all the fold changes, 2 decimals seem more appropriate (for example: for lnc-SAMD14-4, 86.11 seems more appropriate than 86.11047, etc...).

Figure 1: why do you show the volcano plot with the lncRNAs and mRNAs combined, whereas the heatmaps show lncRNAs and mRNAs separately?

Figure 3 legend in the pdf file: H and I should probably be A and B (the word document has been correctly modified).

L204: “Fu’s group… and Liu’s group…compared with controls”: could you insert the references at the end of this sentence?

---

## Round 0.5 · Minor Revisions

Your article is now scientifically acceptable, however a final check by Amanda Toland, one of the Section Editors for PeerJ, has identified that there are still a number of typos and grammatical errors which should be resolved in a final round of minor revisions.

Regarding the change in address ([email protected]), please contact PeerJ administration.

---

## Round 0.6 · accepted · Accept

The production groupe of PeerJ will perform a last general review looking for English typos.